# ATG101 Degradation by HUWE1-Mediated Ubiquitination Impairs Autophagy and Reduces Survival in Cancer Cells

**DOI:** 10.3390/ijms22179182

**Published:** 2021-08-25

**Authors:** JaeYung Lee, Jiyea Kim, Jeongeun Shin, YongHyun Kang, Jungwon Choi, Heesun Cheong

**Affiliations:** 1Department of Cancer Biomedical Science, Graduate School of Cancer Science & Policy, National Cancer Center, Goyang-si 10408, Korea; 97144@ncc.re.kr (J.L.); wldp510@gmail.com (J.K.); 97142@ncc.re.kr (J.S.); 97161@ncc.re.kr (Y.K.); 2Division of Cancer Biology, Research Institute, National Cancer Center, Goyang-si 10408, Korea; 75876@ncc.re.kr

**Keywords:** autophagy, mitophagy, cancer, autophagy-related gene 101(ATG101), Unc-51-like kinase 1(ULK1), HECT, UBA and WWE domain containing E3 ubiquitin protein ligase 1(HUWE1), E3 ubiquitin ligase, ubiquitination, WD repeat domain, phosphoinositide interacting 2 (WIPI2)

## Abstract

Autophagy is a critical cytoprotective mechanism against stress, which is initiated by the protein kinase Unc-51-like kinase 1 (ULK1) complex. Autophagy plays a role in both inhibiting the progression of diseases and facilitating pathogenesis, so it is critical to elucidate the mechanisms regulating individual components of the autophagy machinery under various conditions. Here, we examined whether ULK1 complex component autophagy-related protein 101 (ATG101) is downregulated via ubiquitination, and whether this in turn suppresses autophagy activity in cancer cells. Knockout of ATG101 in cancer cells using CRISPR resulted in severe growth retardation and lower survival under nutrient starvation. Transfection of mutant ATG101 revealed that the C-terminal region is a key domain of ubiquitination, while co-immunoprecipitation and knockdown experiments revealed that HECT, UBA and WWE domain containing E3 ubiquitin protein ligase 1(HUWE1) is a major E3 ubiquitin ligase targeting ATG101. Protein levels of ATG101 was more stable and the related-autophagy activity was higher in HUWE1-depleted cancer cells compared to wild type (WT) controls, indicating that HUWE1-mediated ubiquitination promotes ATG101 degradation. Moreover, enhanced autophagy in HUWE1-depleted cancer cells was reversed by siRNA-mediated ATG101 knockdown. Stable ATG101 level in HUWE1-depleted cells was a strong driver of autophagosome formation similar to upregulation of the known HUWE1 substrate WD repeat domain, phosphoinositide interacting 2 (WIPI2). Cellular survival rates were higher in HUWE1-knockdown cancer cells compared to controls, while concomitant siRNA-mediated ATG101 knockdown tends to increase apoptosis rate. Collectively, these results suggest that HUWE1 normally serves to suppress autophagy by ubiquitinating and triggering degradation of ATG101 and WIPI2, which in turn represses the survival of cancer cells. Accordingly, ATG101-mediated autophagy may play a critical role in overcoming metabolic stress, thereby contributing to the growth, survival, and treatment resistance of certain cancers.

## 1. Introduction

Autophagy is an intracellular catabolic pathway that serves to maintain cellular homeostasis under stress by eliminating deleterious structures (such as misfolded proteins and dysfunctional organelles) through lysosomal degradation and by enhancing the availability of basic nutrient molecules through recycling. Under metabolic stress conditions, such as nutrient deficiency or growth factor deprivation, unnecessary cellular components, including damaged organelles or long-lived proteins, are enwrapped by vesicular membranes and degraded by lysosomal hydrolases, thereby creating auxiliary nutrient pools [1,2,3]. Autophagy plays critical roles in various pathophysiological status, including cancer development as an intracellular survival process [4,5,6]. The entire process of autophagy can be divided into several steps, initiation, nucleation, autophagosome extension, and lysosomal fusion/degradation, each involving distinct molecular complexes and regulated by unique signaling pathways. These distinct steps of regulatory processes have been well-conserved in many organisms from lower eukaryotes to mammals, proved by genetic ablation and mutation analyses.

For autophagy initiation, Unc51-like kinase 1 (ULK1), the mammalian homologue of the first identified autophagy-related gene (ATG), ATG1, forms a complex with three other ATG proteins: ATG13, ATG101/C12orf44, and 200 KDa FAK Family Kinase-Interacting Protein(FIP200)/ RB1 Inducible Coiled-Coil 1 (RB1CC1). In addition to binding for complex formation, ULK1 can also phosphorylate these components, thereby modulating protein function. This ULK1 kinase activity is also regulated by site-specific phosphorylation, with phosphorylation by AMP activated protein kinase (AMPK) promoting and phosphorylation by mammalian target of rapamycin complex 1 (mTORC1) inhibiting kinase activity. AMPK and mTORC1 act as key regulators of autophagy processes at initiation stage through ULK1 phosphorylation at distinct residues [7,8,9]. Through these pathways, autophagy activity is coordinated with metabolism, biosynthesis, and cell growth.

In addition to phosphorylation, multiple components of the autophagy machinery are functionally regulated by other forms of post-translational modification, including ubiquitination, which regulates protein levels by marking proteins for intracellular degradation [10,11,12]. In contrast to ULK1, however, the molecular mechanisms regulating the functional activity and stability of individual ATG proteins involved in distinct steps of autophagy are still largely unknown.

The ULK1 complex core component ATG101 was originally identified as an interacting partner of ATG13 that stabilizes ATG13 within the ULK1 complex. Recent mutation studies using the GFP-LC3 (Microtubule-associated proteins 1A/1B light chain 3B; LC3) puncta assay as an index of autophagy activity have revealed structural features of ATG101 relevant to ATG13 binding and other function, including recruitment of downstream factors to the autophagosome formation site via a WF finger domain [13,14,15]. Our recent mutation studies have also identified the protruding C-terminal domain of ATG101 as a key structure for interacting with multiple class III phosphatidylinositol 3-kinase (PtdIns3K) complex components such as Beclin1, Atg14, and Vps34 [16]. Therefore, ATG101 plays a critical role in linking ULK1 and PtdIns3K complexes for activation of autophagy pathways.

Autophagy and the ubiquitin proteasome system (UPS) are the two major cellular degradation pathways and both are critical for maintaining cellular homeostasis [17]. Further, UPS and autophagy are closely associated and coordinated via multiple signaling pathways. Ubiquitination targets misfolded proteins, protein aggregates, and malfunctioning organelles for degradation and component recycling. In addition, the function and fate of intact proteins can be determined by the type of ubiquitination [12,17,18]. Ubiquitination of ATG proteins regulates their functional activities either positively or negatively, depending on specific types of ubiquitin chain for conjugation. Context-dependent levels of autophagy activity depend on the precise regulation of ATG protein levels [10]. Therefore, the individual steps of autophagy can be regulated via various ubiquitination reactions that control distinct ATG protein levels.

The ULK1 complex is responsible for autophagy initiation, and the activities of complex constituents are regulated by both phosphorylation and ubiquitination. ULK1 ubiquitination is mediated by various autophagy proteins and E3 ligases, including the AMBRA1–TRAF6 complex, chaperone-like protein p32, and Cul3-KLHL20 ubiquitin ligase. Different E3 ligases play distinct roles in regulating ULK1 activity by conjugating specific types of ubiquitin chains. For instance, TRAF6 E3 ligase positively regulates ULK1 activity by conjugating K63 ubiquitin chains. This association is controlled by the molecular mediator AMBRA1 and depends on ULK1 phosphorylation status [19]. In addition, a chaperone-like protein named p32 contributes to K63-linked ubiquitination of ULK1, resulting in greater stability and maintenance of autophagy activity [20]. Conversely, different types of E3 ligases ubiquitinate ULK1 for its degradation. For instance, the Cul3-KLHL20 complex ubiquitinates phosphorylated ULK1 by adding K48 ubiquitin chains, which marks ULK1 for degradation and thereby terminates autophagy [21]. NEDD4-like E3 ligase (NEDD4L) plays a pivotal role in the ubiquitination of ULK1, which disrupts its stability leading to degradation [22]. Critically, this regulation has important implications for cancer progression and treatment response. Ubiquitination by NEDD4-like E3 ligase (NEDD4L) in pancreatic cancer cells promoted ULK1 degradation, while NEDD4L depletion activated autophagy by stabilizing ULK1, which in turn supported cancer progression and survival [23].

While the effects of ULK1 and Beclin1 ubiquitination on autophagy and cell viability have been investigated extensively, relatively little is known about the effects of specific ubiquitination reactions on other components of the ULK1 complex, such as ATG101. Here we investigated novel roles for ATG101 ubiquitination in modulating autophagy activity and cancer cell survival. We demonstrate ATG101 ubiquitination by K48-linked ubiquitin chains mediated by the E3 ubiquitin ligase HUWE1 and regulated by the ATG101 C-terminal domain, resulting in ATG101 degradation and suppression of autophagy. This in turn promoted cancer cell death under stress. Thus, stabilization of ATG101 is critical for the control of autophagy activity and, thus, may be a major determinant of survival during metabolic stress.

## 2. Results

### 2.1. Autophagy-Related Protein 101 (ATG101) Supports Cancer Cell Growth and Survival Via Autophagy

To examine whether ATG101 influences cancer growth and survival via induction of autophagy, we first conducted assays comparing proliferation rates of MIA PaCa-2 pancreatic cancer cells between sgControl (irrelevant sgRNA)-transfected cells and corresponding sgATG101-transfected ATG101 knockout (KO) cells generated using the Lenti-CRISPR system. Compared to corresponding sgControl cell lines, ATG101 KO cell line demonstrated slower cell proliferation rate, even under nutrient complete conditions, as measured using the image-based IncuCyte™ system (Figure 1a). Annexin V/propidium iodide (PI) staining and fluorescence activated cell sorting (FACS) also revealed greater portions of dead cells (Annexin V- or PI-positive) among ATG101 KO cells compared to WT MIA PaCa-2 cells, under nutrient deprivation (Figure 1b). Thus, ATG101 appears to promote the proliferation and survival of cancer cells, possible by activating or maintaining autophagy, a notion investigated in subsequent experiments.

To directly examine the contribution of impaired autophagy from ATG101 KO to reduced cancer cell proliferation and survival, we measured autophagy activity under nutrient starvation by immunoblotting and subcellular localization of the autophagy marker proteins. Immunoblot analysis revealed more accumulated levels of p62 and cytoplasmic LC3-I form than the levels of autophagosome-bound LC3-II forms (lower levels of the ratio LC3-II/I) in ATG101 KO cancer cells compared to corresponding sgCTL cells under both nutrient complete and amino acid-deprived conditions (–AA) (Figure 1c). Moreover, ATG101 KO cells stably expressing GFP-tagged LC3 showed significantly fewer GFP-LC3 puncta (measured as the ratio of the green fluorescent puncta area of GFP to total DAPI stained area) compared to sgControl cells under both nutrient complete and amino acid-deprived conditions (Figure 1d). These results indicate that ATG101 has a critical role on autophagy induced by nutrient starvation.

Based on previous findings implicating ULK1 in mitochondria clearance [24], we hypothesized that the ULK1 complex component ATG101 is also required for mitophagy. Mitophagy—a type of selective autophagy—can be induced by carbonyl cyanide m-chlorophenyl hydrazine (CCCP), a potent mitochondrial oxidative phosphorylation uncoupler [25], and monitored by mito (mt)-Keima, a type of mitophagy marker. To examine whether ATG101 depletion reduces mitophagy activity, we utilized the pH-dependent dual-color fluorescence reporter mito (mt)-Keima to monitor the rate of mitochondrial degradation. HeLa cells stably expressing mt-Keima together with Parkin, exhibited a rise in red fluorescent signal during CCCP treatment, confirming that these signals represent mitophagy, a lysosomal mitochondrial degradation. Further, mt-Keima red signals were substantially reduced by transfection of a small interfering (si)RNA targeting ATG101 (siATG101) compared to cells transfected with a control siRNA (siControl) (Appendix A). Overall, these findings indicate that ATG101 is a key regulator of autophagosome formation in both nonselective and selective autophagy processes.

### 2.2. ATG101 Is Ubiquitinated through C-Terminal Region

Previously, we (and others) have reported that ULK1 is degraded through poly-ubiquitination [19,20,23,26], suggesting that the UPS is an important regulator of autophagy under certain conditions. Further, we found that regulation of ULK1 by ubiquitination in multiple cancer cell lines and an in vivo model of pancreatic cancer altered cancer proliferation and survival [23]. Here, we examined whether the ULK1 complex component ATG101 is also regulated by ubiquitination, and conducted additional experiments to identify the types of ubiquitin-conjugation and enzymes involved, as such information could help define therapeutic targets for autophagy modulation in cancer.

Similar to ULK1, ATG101 in the same complex for autophagy initiation, the protein levels were gradually reduced in MIA PaCa-2 and HeLa cancer cell lines by treatment with the protein synthesis inhibitor cycloheximide (CHX). To examine whether this reduced expression was mediated by proteasomal or lysosomal degradation, we monitored protein levels during treatment with the proteasome inhibitor MG132 or the lysosomal inhibitor chloroquine (CQ). Only MG132 treatment substantially preserved ATG101 levels, suggesting that ATG101 is degraded primarily via the proteasomal pathway (Figure 2a). To investigate the contribution of ubiquitination to proteasomal degradation, we performed co-immunoprecipitation (co-IP) analysis using HA-ubiquitin and FLAG-ATG101. Subsequent western blotting revealed that immunoprecipitated ATG101 included a larger ubiquitinated form (Figure 2b), which showed similar levels in both nutrient complete- and deprived conditions (Appendix A).

Previous functional and structural studies have suggested that the C-terminal region of ATG101 is required for autophagy activity. It has also been reported that the ATG101 C-terminal region is necessary for interactions with Class III PI3K complexes and maintenance of autophagy activity [16] (Appendix A). To determine the C-terminal region as a ubiquitination target, we transfected ATG101 KO cells with an ATG101 construct deleting the 20-amino acid C-terminal region (ATG101∆C) (Appendix A). On western blot analysis, transfected ATG101∆C was accumulated more in CHX-treated ATG101 KO cells than exogenous full length ATG101 (FL) (Figure 2c).

To provide further support for C-terminal regulation of ATG101 ubiquitination, we performed additional co-IP experiments on ATG101 KO cells co-expressing exogenous ATG101∆C or ATG101 FL with HA-Ubiquitin. Co-immunoprecipitation revealed substantial ubiquitination of ATG101 FL, but little ATG101∆C ubiquitination (Figure 2d). Moreover, exogenous ATG101∆C protein levels decreased more slowly under CHX treatment than exogenous ATG101 FL levels in ATG101 KO cells (Figure 2e).

Given that the types of conjugated ubiquitin are major determinants of target protein function and fate, we next examined the structures of ubiquitin chains conjugated to ATG101 by co-IP of ATG101 with distinct WT ubiquitin, Lys-27-only, Lys-48-only, and Lys-63-only ubiquitin constructs in which all other lysine residues were replaced by arginine. The levels of ubiquitin-conjugated ATG101 were monitored by pull-down analysis of FLAG-ATG101 with these specific lysine-only ubiquitin mutants and subsequent western blotting. The immunoprecipitated protein complexes of HA-Ub showed overall similar ubiquitin levels among various ubiquitin mutants. However, co-immunoprecipitation of the Lys-48-linked ubiquitin revealed lower ATG101 levels rather than immunoprecipitated complex with WT ubiquitin-, Lys-27- or Lys-63-linked ubiquitin mutant, implying that Lys-48 ubiquitin may be the ubiquitin type of link to ATG101 (Figure 2f).

### 2.3. ATG101 Single Lysine Mutant in C-Terminal Region Is not Sufficient for Altering Autophagy Activity

To identify the specific residues ubiquitinated in ATG101, we first generated an ATG101 mutant in which K213, the sole lysine residue within the 20-amino acid C-terminal region (amino acids 199–218), was replaced by arginine (K213R) (Appendix A). Then, the stability and ubiquitination status of ATG101 K213R was compared to ATG101 FL by co-IP analysis and western blotting. Protein levels of the ATG101 K213R mutant remained relatively stable under CHX treatment compared to WT ATG101 (Appendix A), indicating that ATG101 K213R may be a potential ubiquitination site triggering ATG101 degradation. However, western blotting subsequent to co-IP assays with FLAG-ATG101 K213R or WT FLAG-ATG101 and HA-Ubiquitin revealed similar levels of ubiquitin (Appendix A), implying that K213 is not the only site for ATG101 ubiquitination. As part of the ULK1 complex, ATG101 is required for autophagy initiation. To examine regulation of autophagy by K213 ubiquitination, we first investigated if the ATG101 K213R mutant could rescue autophagy in ATG101 KO cells under nutrient starvation. Unexpectedly, however, GFP-LC3 puncta assays revealed that ATG101 K213R could not restore autophagy as effectively as WT ATG101 (Appendix A). These results indicate that the single lysine mutation in the C-terminus is not sufficient to alter total ubiquitination, although the stability of ATG101 is influenced by this mutation.

### 2.4. HUWE1 Mediates ATG101 Ubiquitination and Degradation

To identify upstream E3 ubiquitin ligases targeting ATG101, we first screened for ATG101-interacting proteins in MIA PaCa-2 pancreatic cancer cells using immunoprecipitation and mass spectrometry. The E3 ubiquitin ligases immunoprecipitated with ATG101 are listed in Appendix A. Among these E3 candidates, we examined whether HUWE1 regulates ATG101 stability, because HUWE1 has been reported to ubiquitinate and trigger the degradation of the autophagy protein WIPI2, thereby impacting autophagy activity [27]. Consistent with a similar function, MIA PaCa-2 cells transfected with a HUWE1-targeted siRNA exhibited relatively higher ATG101 protein levels than cells transfected with siControl (siCTL) under CHX treatment (Figure 3a).

We next investigated whether HUWE1 can influence ATG101 stability by regulating its ubiquitination. Cells were transfected with FLAG-ATG101 and HA-Ubiquitin after transfection with several siRNAs targeting E3 ligases, and ubiquitin measured by western blotting. Cells transfected with siHUWE1 yielded significantly lower ubiquitin levels compared to cells transfected with siCTL or siRNAs targeting other E3 ligases, including TRAF6 (Figure 3b). We also performed a reverse immunoprecipitation analysis with HA-Ub in shCTL and shHUWE1 cells after co-transfection with FLAG-ATG101. Then ATG101 levels from the immunoprecipitated complexes of HA-Ub were analyzed by western blotting. ATG101 levels pulled down with ubiquitin in shHUWE1 cells revealed significantly lower levels compared to that in shCTL cells (Figure 3c), suggesting that HUWE1 is the predominant E3 ligase catalyzing ubiquitination of ATG101. Interestingly, ubiquitination levels of ATG101 in nutrient-deprived conditions were comparable to that in nutrient-complete conditions based on the immuno-blot analysis of ubiquitin levels from the immunoprecipitated complexes of FLAG-ATG101 (Appendix A).

Moreover, we also investigated the physical interaction between ATG101 and HUWE1 using co-IP and found that endogenous HUWE1 co-immunoprecipitated with endogenous ATG101 in MIA PaCa-2 pancreatic cancer cells as well as with ectopically expressed FLAG-ATG101 in HEK293T cells using a primary antibody against HUWE1, confirming a physical association between ATG101 and HUWE1 in basal conditions (Figure 3d,f). The interaction of ATG101 and HUWE1 under the nutrient starvation conditions was similar to that observed in the basal conditions (Figure 3e,f). Collectively, these results suggest that HUWE1 is the primary E3 ligase targeting ATG101 and that HUWE1-mediated ubiquitination negatively regulates ATG101 protein stability.

### 2.5. HUWE1 Suppresses Autophagy Activity through Regulating ATG101 Levels

Next, we examined whether HUWE1-mediated ATG101 degradation regulates autophagy activity. In cells expressing GFP-LC3, stable HUWE1 knockdown using a short hairpin RNA (shRNA) substantially enhanced GFP-LC3 puncta formation (a marker for autophagosomes) compared to GFP-LC3-expressing cells transfected with a shCTL, even under nutrient complete conditions (Figure 4a). The GFP component of the GFP-LC3 fusion protein can be released during lysosomal degradation, and the rate of free GFP accumulation is usually enhanced under autophagy-activating conditions, such as nutrient deprivation or treatment with the mTOR inhibitor rapamycin [28]. Free GFP levels from GFP-LC3 and p62 degradation rate were higher following shHUWE1 transfection compared to shCTL transfection (Figure 4b), indicating that HUWE1 depletion substantially increased basal autophagy activity. However, when ATG101 was also knocked down, the enhancement of GFP-LC3 puncta formation in shHUWE1 cells was reduced to a level similar to that observed in shCTL-transfected cells, particularly under treatment with rapamycin (Figure 4c; Appendix A), suggesting that ATG101 protein levels and autophagy are negatively regulated by HUWE1.

HUWE1 was also reported to control WIPI2 protein levels by regulating ubiquitination [27]; we then examined possible molecular associations between WIPI2 and ATG101 under HUWE1 regulation. When mCherry-GFP-LC3 expressing cells was utilized for monitoring autophagy flux, WIPI2 knockdown by siRNA in shHUWE1 cells also reduce mCherry-LC3 puncta formation occurred in lysosome to the same extent as in cells transfected with siATG101 in shHUWE1 cells (Figure 4d; Appendix A). However, when WIPI2, a known substrate of HUWE1, was knocked down by siRNA upon shHUWE1 cells, mCherry-GFP-LC3 puncta formation in siWIPI2 was similar defect as that of siATG101 cells by showing a significant reduction of mCherry puncta formation (Figure 4d; Appendix A). Moreover, mCherry-LC3 puncta formation in both WIPI2/ATG101 double knockdown cells was comparable to that in either ATG101 or WIPI2 single knockdown cells (Figure 4d; Appendix A), implying that autophagic flux shown by mCherry-LC3 puncta formation requires either ATG101 or WIPI2.

Given that WIPI2 and ATG101 share a common upstream ligase despite acting in different phases of autophagy, we tested for potential physical interaction between WIPI2 and ATG101. Immunoprecipitation with anti-GFP after co-transfection of FLAG-ATG101 with GFP-WIPI2 showed a band corresponding to the FLAG-ATG101, which co-immuno-precipitated with the GFP-WIPI2 (Figure 4e), indicating that ATG101 binds to WIPI2 in the presence or absence of rapamycin treatment. Moreover, when we performed immunoprecipitation analysis of either FLAG-ATG101 (FL) or C-terminus deleted mutant (ΔC) with GFP-WIPI2, the bands corresponding to the GFP-WIPI2 that co-immunoprecipitated with the FLAG-ATG101, were significantly weaker levels with ATG101ΔC rather than those with ATG101 FL. These results suggest that the ubiquitination of ATG101 might impact on the interaction of ATG101 and other downstream target, WIPI2 (Figure 4f).

### 2.6. ATG101 Ubiquitination Is Critical for Supporting Cancer Cell Survival under Metabolic Stress Conditions

Knockout of ATG101 markedly reduced the proliferation rate and survival of cancer cells, especially under nutrient deprivation (Figure 1a,b), strongly suggesting that autophagy is a critical survival mechanism for these cells under metabolic stress. These findings also suggest that HUWE1-mediated ATG101 degradation may reduce cancer cell survival. To investigate this possibility, we compared survival rates between cells stably transfected with shCTL or shHUWE1, a treatment already demonstrated to enhance autophagy under both nutrient complete and deprivation conditions (Figure 4a,b), by Annexin V/PI staining and FACS analysis. Consistent with a protective function for autophagy, the rate of apoptotic cell death was substantially lower in cells transfected with shHUWE1 compared to cells transfected with shCTL, potentially due to stabilization of ATG101 (Figure 5a). Consistent with this notion, additional deletion of ATG101 in shHUWE1-transfected cells increased apoptotic death rate compared to shHUWE1 single knockdown. Moreover, overall apoptotic death rate was higher in shHUWE1 knockdown cells with siRNA-mediated depletion of both WIPI2 and ATG101 compared to shHUWE1 knockdown cells suggesting that these molecules more likely are stabilized in HUWE1 depleted conditions and act cooperatively to sustain autophagy (Figure 5b). Overall, our data indicates that modulation of ATG101 protein levels by HUWE1-mediated ubiquitination and ensuing changes in autophagy activity influence the survival of cancer cells under metabolic stress (Figure 5c). These findings may have important implications for cancer treatment as they suggest that concomitant suppression of autophagy will enhance the therapeutic efficacy of anticancer agents.

## 3. Discussion

Recent studies have identified various molecular mechanisms regulating ULK1 protein levels. For example, multiple E3 ligases such as TRAF6 and NEDD4L regulate ULK1 stability through ubiquitination and other post-translational modifications such as phosphorylation and glycosylation [17,18,19,20,21,22,23]. However, relatively little is known about the regulation of the other ULK1 complex components ATG13 and ATG101.

Here, we describe a series of gene-knockout/knockdown, subcellular localization of autophagy markers, and co-IP experiments demonstrating that ATG101 levels are regulated by HUWE1-mediated ubiquitination and subsequent proteasomal degradation in cancer cells and this degradation pathway can markedly suppress autophagy, leading to reduced cell viability under metabolic stress. Thus, activation of this HUWE1/ATG101 pathway may be a feasible clinical strategy to impair cancer cell survival or enhance the efficacy of anti-tumor therapies.

ATG101 (C12orf44) was originally identified as a scaffold protein that maintains ATG13 stability within the ULK1 complex [13,14]. Recent structural studies have revealed a highly structured Hop1, Rev7, Mad2 (HORMA) domain of ATG101 is involved in the interaction with ATG13-ULK1 and in recruiting downstream proteins to the autophagosome site [15]. In addition, ATG101 includes a flexible C-terminal domain that mediates interactions with class III PI3K complex (PtdIns3K) proteins, such as Vps34, Beclin1, and ATG14, resulting in autophagy activation [16] (Appendix A). In contrast, the Hedgehog (Hh) receptor Patched1 (PTCH1) was shown to bind ATG101 through the PTCH1 C-terminal domain and inhibit autophagy flux, which further influences PTCH1-dependent tumor suppression independent of Sonic Hedgehog canonical signaling [29]. More recently, additional physiological functions of ATG101 have been revealed by analysis of an Atg101 loss-of-function mutant fly. In this model, Atg101-mediated autophagy maintains neural and midgut homeostasis and further influences adult lifespans [30].

The functions of ATG101 may also be regulated by ubiquitination, a key post-translational modification regulating the expression levels and activities of many proteins. We found that ATG101 is degraded through E3 ligase-driven poly-ubiquitination, likely involving the C-terminal region of ATG101 as evidenced by altered ubiquitination of a C-terminal deletion mutant (ATG101∆C) (Figure 2c,d). Further, this ubiquitination markedly suppressed ATG101 protein levels, thereby reducing autophagy activity and cell viability.

Given that ubiquitin types of conjugation determines function of the target proteins, we examined which specific types of ubiquitin are added onto ATG101 for modulating targets. The fate of an ubiquitinated protein is determined by the specific conjugation pattern. Multiple reports have suggested that Lys-63-linked ubiquitin chains target proteins for autophagic degradation [31], whereas proteins conjugated to Lys-48- or Lys-27-linked chains are likely to undergo proteasomal degradation [32]. In this study, we found preferential conjugation of K48-only ubiquitin mutant to the ATG101 upon immunoprecipitation analysis (Figure 2f), and pharmacological inhibition experiments suggested that this ubiquitination process leads to ATG101 elimination via a proteasomal pathway (Figure 2a).

HUWE1 was identified as an upstream E3 ligase catalyzing ATG101 ubiquitination in cancer cell lines based on the substantial reduction in ubiquitinated ATG101 upon HUWE1 KD (Figure 3b). In addition, we provide evidence for a potential physical interaction between HUWE1 and ATG101 through immunoprecipitation analysis (Figure 3c). We also found that reduced ubiquitination by HUWE1 KD led to ATG101 accumulation (Figure 3a), further supporting ubiquitination-induced proteasomal degradation as an important regulatory mechanism for maintaining ATG101 levels. Moreover, knockdown of HUWE1 enhanced the rate of LC3 puncta formation in cancer cells under metabolic stress (Figure 4a), indicating that this regulatory mechanism for ATG101 levels directly modulates autophagy.

HUWE1 is an E3 ubiquitin ligase harboring HECT domain, which is known to regulate cell proliferation and cell death and, thus, is potentially an important factor in tumorigenesis [33,34,35,36]. However, functional studies have reported discordant effects of changes in HUWE1 activity due to its functionally diverse targets. For instance, HUWE1 substrates include both anti- and pro-apoptotic factors [35,37,38]. HUWE1 has also demonstrated dual roles during tumorigenesis, again reflecting the functional diversity of target substrates, including both oncogenic molecules, such as c-Myc and MIZ-1[39,40], as well as tumor suppressing molecules such as P53 [38] and BRCA1 [41,42]. A recent report identified the autophagic protein WIPI2 as another potential substrate for ubiquitination by HUWE1. Further, this regulation was negatively regulated by mTORC1-mediated phosphorylation [27]. WIPI2 was identified to contribute to the autophagosome elongation step by recruiting ATG 12-5-16 complexes to the autophagosome precursor forms [43]. Accordingly, we examined whether increased autophagy in HUWE1-depleted cancer cells could be suppressed by knockdown of WIPI2 as well as by ATG101 knockdown. Indeed, GFP-LC3 puncta formation, a marker for autophagosomes, was enhanced by shHUWE1 cells and suppressed significantly by an siRNA against ATG101 (Figure 4c). However, both siATG101 and siWIPI2 transfection showed similar suppression compared to that of single knockdown, siWIPI2 or siATG101 (Figure 4d), implying that both ATG101 and WIPI2 act as a critical role on the same pathway for regulating autophagy activity, despite being involving in distinct stages. Interestingly, the physical association between ATG101 and WIPI2 is mediated through the C-terminal domain of ATG101, which is highly ubiquitinated (Figure 4e.f).

Finally, we demonstrated that ATG101-mediated autophagy facilitated while ATG101 downregulation by HUWE1-mediated ubiquitination impaired cancer cell survival (Figure 5a). Further, double knockdown of both ATG101 and WIPI2 in shHUWE1 cancer cells significantly increased the apoptotic death rate compared to that in shCTL cells (Figure 5b). Thus, either the HUWE1/ATG101 or HUWE1/WIPI2 pathway could be potential targets for suppressing tumor cell survival, and these reverse combinational approaches may be more effective.

Our results demonstrate that HUWE1 destabilizes ATG101 by poly-ubiquitination at the C-terminus, thereby suppressing ATG101-mediated autophagy activity and further inhibiting cancer cell survival under nutrient deprivation conditions (Figure 5c). Due to this reciprocal regulation of ATG101 and HUWE1, ATG101-mediated autophagy activation under HUWE1 depletion may overcome the metabolic stressors frequently encountered by cancer cells. As discussed earlier, suppression of autophagy may be a feasible strategy to limit tumor growth or enhance anticancer treatment efficacy. However, the precise functions of HUWE1 in cancer progression are controversial, as they are determined not only by the physiological functions of target substrates, but also by other types of post-translational modifications. Therefore, additional studies, including in vivo experiments using spontaneous cancer mouse models, are needed to better understand the tumor suppressing functions of HUWE1 mediated by elimination of autophagy proteins. Based on our findings, however, further studies are warranted on the contributions of HUWE1-mediated autophagic protein ubiquitination to cancer progression and treatment response.

## 4. Materials and Methods

### 4.1. Cell Lines

MIA PaCa-2, HEK293T, and HeLa cells were kindly provided by YH Kim, KT Kim (National Cancer Center Korea), which are originally purchased from the American Type Culture Collection (ATCC; Manassas, VA, USA). mt-Keima and Parkin stably expressing HeLa cells was kindly provided by Dr. Jeanho Yun (Dong-A University, Busan, Korea). All cells were maintained at 5% CO_2_ and 37 °C in Dulbecco’s Modified Eagle Medium (DMEM) supplemented with 10% fetal bovine serum (FBS; HyClone, Logan, UT, USA), 100 U/mL penicillin, and 100 μg/mL of streptomycin (Gibco, Waltham, MA, USA) For starvation media, Earle’s Balanced Salt Solution (EBSS) or Hank’s balanced saline solution (HBSS) was used as a base solution and then supplemented with 10% dialyzed FBS, glucose, vitamins, HEPES and minerals at the same concentration as in DMEM.

### 4.2. Generation of Stable Cell Lines

GFP-LC3 or mCherry-GFP-LC3 was stably expressed in HeLa and MIA PaCa-2 cells using a retroviral vector following standard protocols for viral transduction. For generating HUWE1 knockdown cell lines, Lenti-viral vector (pLKO.1; Addgene) expressing shRNA against HUWE1 was constructed. The following shRNA sequences were used for the constructs : Forward : 5′-CCGGCCACACTTTCACAGATACTATCTCGAG ATAGTATCTGTGAAAGTGTGGTTTTTG-3′, Reverse: 5′- AATTCAAAAACCACACTT

TCACAGATACTATCTCGAGATAGTATCTGTGAAAGTGTGG -3′. Stable knockdown cells were generated using the lentiviral vector harboring either shHUWE1 or a scramble shRNA as a control following standard protocols for viral transduction.

For generating non-targeting sgRNA and ATG101 KO cell lines, LentiCRISPRv2-based ATG101 CRISPR-Cas9 guide RNA expression plasmid (Gene Script, U0448BI200-1; Piscataway, NJ, USA,) and LentiCRISPRv2-sgControl expression plasmid was used. The following sgControl sequence was used for the constructs : 5′-CACCGGCACTACCAGAGCTAACTCA-3′. Then viral transduction processes were followed by standard protocols. Subsequently after appropriate selection steps, immunoblotting was performed to test the expression of the proper gene sets in stable cell lines.

### 4.3. Antibodies and Reagents

Primary antibodies against ATG13 (13468), ATG101 (13492), LC3B (2775) and WIPI2 (8567) were purchased from Cell Signaling Technology (Danvers, MA, USA); Antibodies against HUWE1 (A300-486), β-actin (A300–491A), HA (A190–108A) and WIPI2 (A305-324A) were purchased from Bethyl Laboratories(Montgomery, TX, USA); those against FLAG M2 (F1804) and FLAG (F7425) were purchased from Sigma Aldrich (St. Louis, MO, USA); antibody against GFP (mouse SC-9996)(rabbit SC 8334) were purchased from Santa Cruz Biotechnology; and antibody against p62(610832) was purchased from BD Bioscience; antibody against ATG14 (GTX119950) was purchased from GeneTex (Hsinchu, Taiwan). Secondary antibodies against horseradish peroxidase-linked anti-rabbit (A120–101P) and anti-mouse (A90–116P), were purchased from Bethyl Laboratories.

Hoechst 33342 (H3570), Lipofectamine^TM^ 2000 (11668019), Lipofectamine^TM^ RNAiMAX (13778150) were purchased from Thermo Fisher Scientific (Waltham, MA, USA). Rapamycin (Rapamycin from *Streptomyces hygroscopicus*, R0395), CCCP (carbonyl cyanide 3-chlorophenylhydrazone, C2759), Chloroquine (CQ, C6628), MG132 (M7449), and Cycloheximide (CHX, C4859) were purchased from Sigma Aldrich. Protease inhibitor cocktail tablets (11697498001) were purchased from Roche Applied Bioscience (Penzberg, Germany).

### 4.4. DNA Construct and siRNA

For constructing the FLAG-ATG101, ATG101 encoding DNA fragment that amplified by a polymerase chain reaction (PCR) was inserted between the EcoRI and XhoI sites of the pCMV9-3x FLAG vector. The full-length ATG101 cDNA was provided by the Korea Human Gene Bank (Daejeon, Korea). A plasmid encoding GFP-LC3B in a MigRI-based retroviral vector removed GFP reporter was generously provided by Dr. Craig Thompson (Memorial Sloan Kettering Cancer Center, New York, NY, USA). A plasmid encoding mCherry-GFP-LC3B in pBabe vector was provided by Dr. Jayanta Debnarth through Addgene (22418) (Watertown, MA, USA). Plasmids encoding HA-ubiquitin and pcDNA3-HA were provided by Dr. Seok Hee Park (Sungkyunkwan University, Seoul, Korea). Plasmids pRK5-HA-ubiquitin-WT, pRK5-HA-ubiquitin-K27, pRK5-HA-ubiquitin-K48, pRK5-HA-ubiquitin-K63 were kindly provided by Dr. Jaewhan Song (Yonsei University, Seoul, Korea).

Negative control siRNA (non-targeting pool) and siRNA targeting the genes of interest were purchased from Genolution Inc. (Seoul, Korea).As following siRNA sequences were used for the indicated target genes: siControl : 5′-CUCGUGCCGUUCCAUCAGGUAGUU-3′ ; siATG101, 5′-ACUUCAUCGACUUCACUUATT-3′ (#1) and 5′-CAGCCCUACCUGUACAAGATT-3′ (#2); siHUWE1, 5′-CAUUGGAAAGUGCGAGUUA-3′ (#1) and 5′-CUGUGAGAGUGAUCGGGAA-3′ (#2); siCHIP, 5′-CGAGCGCGCAGGAGCTCAA-3′ (#1) and 5′-AGCTGGAGATGGAGAGCTA-3′(#2); siTRAF6, 5′-CCACGAAGAGAUAAUGGAUGCCAAA-3′ (#1) and 5′-GTTCATAGTTTGAGCGTTA-3′ (#2); siWIPI2, 5′-TACGGAAGATGTGTGCATT-3′ (#1) and 5′-GACAGUCCUUUAGCGGCATT-3′ (#2).

### 4.5. Mutagenesis

All mutants of ATG101 were generated by site-directed mutagenesis, substituting the central 1–2 nucleotides of the desired mutagenic site with two complimentary mutagenic primers using the Muta-Direct site-directed mutagenesis kit (iNtRON Biotech, Sungnam Korea; cat no. 15071), following the manufacturer’s instructions.

### 4.6. LC–MS/MS Analysis

The protein samples were precipitated using cold acetone, reduced with 10 mM dithiothreitol (DTT), and alkylated with iodoacetamide (IAA). The alkylated samples were digested with mass spec grade trypsin/lys-C mix in 50 mM Tris-HCl (pH 8) for 12 h at 37 °C. The digested peptides were analyzed by a Q Exactive hybrid quadrupole-orbitrap mass spectrometer (Thermo Fisher Scientific) coupled with an Ultimate 3000 RSLCnano system (Thermo Fisher Scientific). The peptides were loaded onto trap columns (100 μm × 2 cm) packed with Acclaim PepMap100 C18 resin, separated on an analytical column (EASY-Spray column, 75 μm × 50 cm, Thermo Fisher Scientific), and sprayed into the nano-electrospray ionization source. The Q Exactive Orbitrap mass analyzer was operated in a top ten data-dependent method. Full MS scans were acquired over a range of 300–2000 *m*/*z* with a mass resolution of 70,000 (at 200 *m*/*z*). The automatic gain control target value was 1.0 × 106. The ten most intense peaks with charge state ≥2 were fragmented in the higher-energy collisional dissociation collision cell with normalized collision energy of 30, and tandem mass spectra were acquired in the Orbitrap mass analyzer with a mass resolution of 17,500 at 200 *m*/*z*. Database searching of all raw data files was performed using Proteome Discoverer 2.2 software (Thermo Fisher Scientific). SEQUEST-HT was used for database searching against the Swiss-Prot Homo sapiens database. Database searching against the corresponding reversed database was also performed to evaluate the false discovery rate (FDR) of peptide identification. The database searching parameters included precursor ion mass tolerance 10 ppm, fragment ion mass tolerance 0.08 Da, fixed modification for carbamidomethyl cysteine, and variable modifications for methionine oxidation. We obtained an FDR of less than 1% on the peptide level and filtered for high peptide confidence.

### 4.7. Immunoprecipitation

Ubiquitin, ATG101 and WIPI2 were tagged with a human influenza HA epitope, FLAG and GFP, respectively. Epitope-tagged proteins were co-expressed in HEK293T and MIA PaCa-2 cells. In the stage of cell harvest, HEK293T cells were washed with ice-cold PBS and lysed in lysis buffer containing 1% NP-40, 0.2 mM PMSF, 10 mM NaF, 20 mM Tris-HCl, 10% glycerol, 2 mM EDTA, 1 mM Na3O4V, 150 mM NaCl, protease inhibitor cocktail (11836153001; Roche Applied Bioscience) and 1% phosphatase inhibitor cocktail (Sigma Aldrich). Then, each 0.5 mg of cell lysates were incubated with 2 μg primary antibodies against FLAG M2 (Sigma Aldrich), HA (Bethyl Laboratories), GFP (Santa Cruz), HUWE1 (Bethyl Laboratories), rabbit IgG (Sigma Aldrich), or mouse IgG (Sigma Aldrich) at 4 °C for 90 min. Then, 50 μL protein A agarose beads (GenDEPOT, Katy, TX, USA) were added and incubated at 4 °C for overnight. Immunoprecipitates were washed three times with wash buffer and then eluted by boiling in Sodium dodecyl sulfate (SDS) sample buffer with β-mercaptoethanol (β-ME) for 5 min. Then, western blot was performed for immunoblotting immunoprecipitates with the indicated antibodies. Liquid chromatography mass spectrometry (LC–MS) was used for analyzing the immunoprecipitated complex.

### 4.8. Fluorescence Microscopy Analysis of Autophagy

Cell lines stably expressing LC3B tagged with GFP were used for monitoring autophagy activity by confocal fluorescence microscopy. Cells stably expressing the tandem mCherry-GFP-LC3 construct were also used. MIA PaCa-2 and HeLa cells stably expressing GFP-LC3 or mCherry-GFP-LC3 and transfected with either control or HUWE1 shRNA were cultured in a glass-bottomed chamber (Lab-Tek; Thermo Fisher Scientific) overnight, and then replaced with DMEM culture medium containing the indicated chemicals or starvation media for the indicated time periods. Nuclei were stained using Hoescht-33342. Images were acquired with the LSM780 confocal fluorescent microscope (Carl Zeiss, Oberkochen, Germany) and the percent of either GFP-LC3 puncta area or mCherry-LC3 puncta area were normalized to the Hoechst 33342-stained area, which was quantified using ZEN black software (Carl Zeiss). The area of LC3 puncta was counted in five different arbitrary areas from three independent experiments.

### 4.9. Fluorescence Microscopy Analysis of Mitophagy

HeLa cells stably expressing mt-Keima and Parkin were used for monitoring mitophagy activity by confocal fluorescence microscopy [44]. Cells were cultured and reverse transfected with siRNA onto a glass-bottomed chamber (Lab-Tek; Thermo Fisher Scientific) for 48 h, and then replaced with DMEM culture medium containing the indicated chemicals or starvation media for the indicated time periods. Nuclei were stained using Hoescht-33342. Images were acquired with the LSM780 confocal fluorescent microscope (Carl Zeiss, Oberkochen, Germany). Fluorescence of mt-Keima was imaged in two channels via two sequential excitations (458 nm, “green” and 561 nm, “red”, respectively) and using a 570 695 nm emission range. The value of mt-Keima red area was divided by the value of mt-Keima green area and then was normalized to the Hoechst-stained area. The quantification was performed using ZEN black software (Carl Zeiss).

### 4.10. Cell Proliferation and Death Assay

Cell proliferation was measured using the image-based cell proliferation analyzer IncuCyte^TM^ (Essen Instruments, Ann Arbor, MI, USA). Cells were cultured in nutrient-complete DMEM media on multi-well plates overnight and imaged throughout the indicated time period. IncuCyte^TM^ automated cell proliferation detector was used to measure cell proliferation through quantitative kinetic processing metrics derived from time-lapse image acquisition and presented as a percentage of cell confluence over time. Cell viability was determined by Annexin V and PI staining following standard protocols at the indicated time periods (556547, BD Biosciences, San Jose, CA, USA). Cells negative for both Annexin V and PI were considered live cells. The proportion of dead cells was measured based on the number of Annexin V and PI single and both-stained cells. The fluorescence of stained cells was detected using the FACS Verse analyzer (BD Biosciences).

### 4.11. Clonogenic Assay

MIAPaCa-2 shControl or shHUWE1 cancer cells were reverse-transfected with each siRNA in 12-well plates at 400 cells/well in duplicate. Then, the cells were kept at 37 °C in 5% CO_2_ for 24 h, and the culture medium was changed into fresh complete media for 4 days. Colonies were fixed with 3.7% Formaldehyde and were stained with 0.5% crystal violet.

### 4.12. Western Blotting

Cells were harvested in ice-cold RIPA lysis buffer (50 mM Tris-Cl, pH 7.4, 150 mM NaCl, 1% NP-40, 0.5% Na-deoxycholate, 0.1% SDS, 1 mM EDTA) containing protease inhibitor cocktail (Roche Applied Bioscience) and phosphatase inhibitor (Sigma Aldrich). Soluble lysate fractions were isolated by centrifugation at 20,000× *g*, for 20 min at 4 °C and quantified using the Pierce bicinchoninic acid (BCA) Protein Assay kit (Thermo Fisher Scientific). Samples were resolved by SDS polyacrylamide gel electrophoresis using equal concentrations of protein and transferred to polyvinylidene fluoride membranes. The membranes were blocked with 5% skim milk and then probed with the indicated primary and secondary antibodies following standard protocols. Image J software (NIH, Bethesda, MD, USA) was used for quantification of the indicated bands.

### 4.13. Statistical Analyses

Immuno-blotted proteins were quantified and using Image J software version 1.50i (NIH, Bethesda, MD, USA) and normalized by loading control. Data are expressed as the mean ± standard error of the mean, which are from at least three independent experiments. Statistical significance was calculated using Student’s t test in Graph Pad Prism 8. A value of *p* < 0.05 was considered statistically significant (* *p* < 0.05; ** *p* < 0.01 ; *** *p* < 0.001).

## Figures and Tables

**Figure 1 ijms-22-09182-f001:**
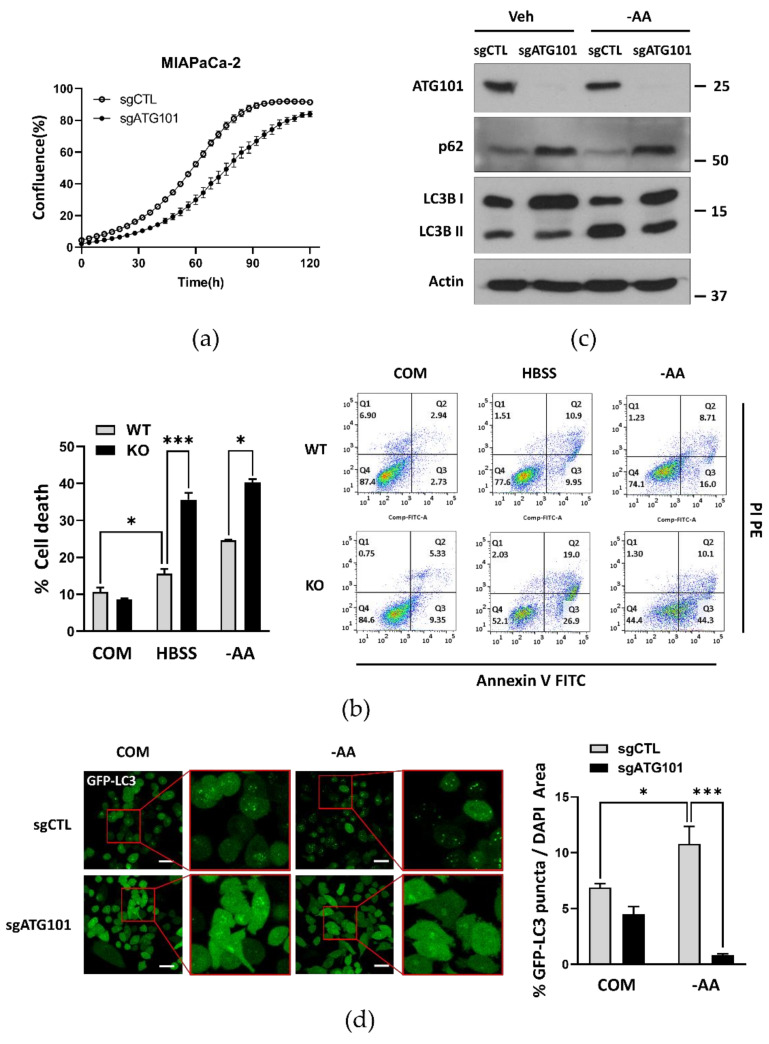
Autophagy-related protein 101 (ATG101) is required for autophagy to promote cell growth and survival. (**a**) Knockout of ATG101 (sgATG101) in MIA PaCa-2 cells using the Lenti-CRISPR system reduced proliferation rate compared to Control (sgCTL) MIA PaCa-2 cells as measured by the IncuCyte^TM^ analyzer. MIA PaCa-2 cells stably expressing GFP-LC3 were used. Cell confluence levels were measured in real-time and expressed as a proportion (%) of complete cell coverage. Error bars indicate the mean ± SEM of three independent experiments. (**b**) Knockout of ATG101 in MIA PaCa-2 cells also enhanced apoptotic death rate under nutrient deprivation as measured by Annexin V/propidium iodide (PI) staining and flow cytometry. WT or ATG101 KO MIA PaCa-2 cells were plated overnight and then incubated in HBSS or the medium without amino acids (amino acid deprivation, −AA) for 48 h. Error bars indicate the mean ± SEM of three independent experiments. * *p* < 0.05; *** *p* < 0.001. (**c**) ATG101 knockout (KO) altered the levels of autophagy markers as measured by western blotting. Cell lysates prepared from Control (sgCTL) or ATG101 KO (sgATG101) stably expressing GFP-LC3 MIA PaCa-2 cells cultured in complete medium(COM) and in amino acid-deprived medium (–AA) for 4 h, were immunoblotted for ATG101, p62, LC3B, and β-actin (gel loading control). (**d**) ATG101 KO reduced the formation of autophagosomes as measured by GFP-LC3 puncta. Control (sgCTL) or ATG101 KO (sgATG101) MIA PaCa-2 cells stably expressing GFP-LC3 were seeded overnight and subsequently starved in amino acid-deprived medium (–AA) for 4 h. Cell nuclei were stained with Hoechst 33342, and images were acquired using a confocal fluorescence microscope. At least five distinct regions were imaged per condition and quantified. Scale bar: 20 µm. GFP-LC3 puncta area was normalized to the Hoechst 33,342 stained area and is presented as a percentage on the quantification graph. Error bars indicate the mean ± SEM of three independent experiments. * *p* < 0.05; *** *p* < 0.001.

**Figure 2 ijms-22-09182-f002:**
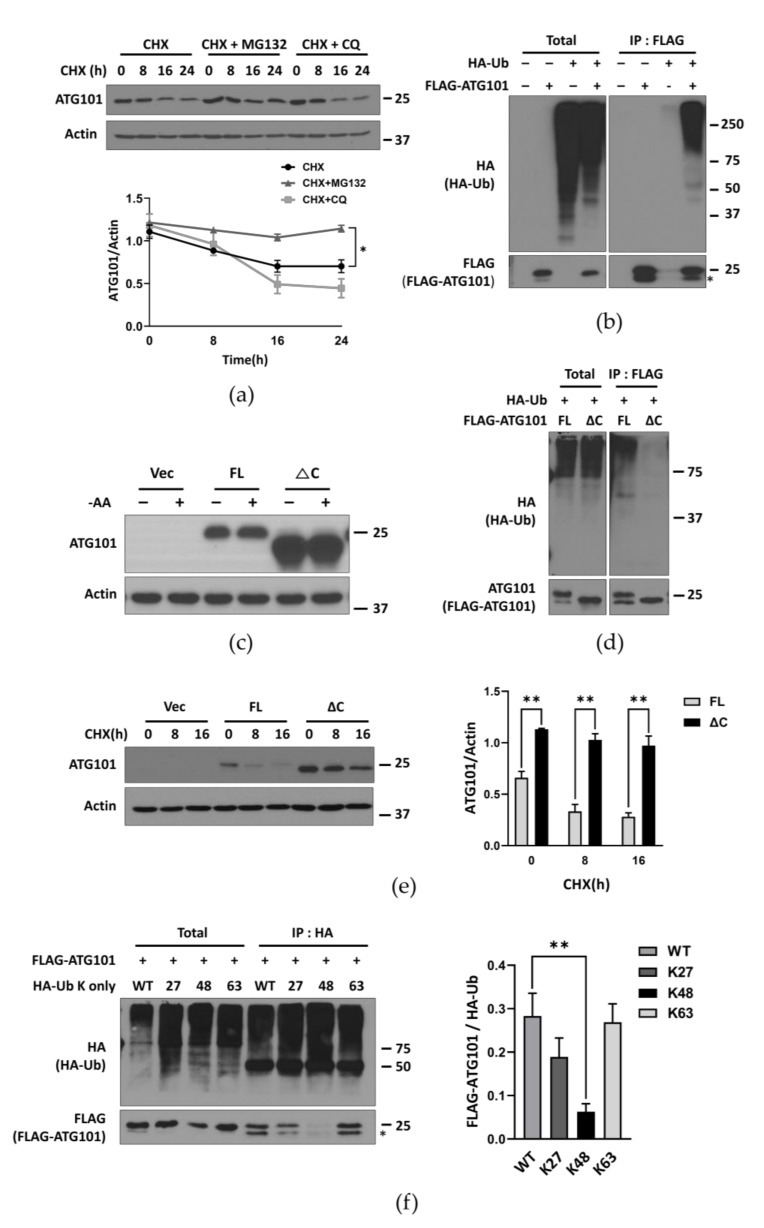
Ubiquitination of ATG101 is regulated through the C-terminal domain. (**a**) Cellular ATG101 is degraded primarily via the proteasomal pathway. MIA PaCa-2 cells were plated overnight and incubated with the protein synthesis inhibitor cycloheximide (CHX, 10 μg/mL) alone or with the proteasome inhibitor MG132 (5 μM) or lysosome inhibitor CQ (10 μM) at the time indicated. For western blot analysis, ATG101 levels were normalized to β-actin as the gel loading control. Western blot band intensities were quantified by Image J. Quantification is presented on the graph. Error bars indicate the mean ± SEM of three independent experiments. * *p* < 0.05. (**b**) Degradation of ATG101 is associated with ubiquitination. ATG101 KO HEK293T cells were transfected with HA-Ubiquitin (Ub) and/or FLAG-ATG101, and FLAG-ATG101 was immunoprecipitated using anti-FLAG. Immunoprecipitates were analyzed by western blot using anti-HA. (**c**) Removal of the ATG101 C-terminus (ΔC) reduced the rate of degradation. ATG101 KO MIA PaCa-2 cell were plated overnight and transfected with FLAG vector (Vec), ATG101 full length (FL) or ATG101ΔC (ΔC) for 24–48 h, then incubated in nutrient complete medium or amino acid deprivation medium (–AA) for 4 h prior to western blot analysis. (**d**) Removal of the ATG101 C-terminus (ΔC) reduced ubiquitination. ATG101 KO HEK293T cells were co-transfected with HA-Ub and FLAG-ATG101 FL or ΔC mutant. FLAG-ATG101 was immunoprecipitated with anti-FLAG, then the immunoprecipitated complexes were analyzed for ATG101 ubiquitination by western blotting using anti-HA. (**e**) Deletion of the C-terminus stabilized ATG101 protein during CHX treatment. MIA PaCa-2 cells were plated overnight, transfected with FLAG Vector (Vec), ATG101 (FL), or ATG101ΔC (ΔC) for 24 h, and then treated with CHX (10 μg/mL). Cell lysates were collected at the indicated time and immunoblotted for ATG101 and β -actin. ATG101 levels were normalized to β-actin (gel loading control). Quantification is presented on the graph. Error bars indicate the mean ± SEM of three independent experiments. ** *p* < 0.01. (**f**) The specific type of ubiquitin tends to conjugate to ATG101. Cells were transfected with ATG101 (FL) and a mutant ubiquitin with only a single lysine (K). The ATG101 promoted conjugation of K48-only ubiquitin. HEK293T cells were transfected with HA-Ub-wild type (WT), K27-only, -K48-only, or -K63-only ubiquitin mutants plus FLAG-ATG101, HA-Ubiquitin was immunoprecipitated using anti-HA, then the immunoprecipitated complexes of HA-Ub were analyzed for ATG101 ubiquitination by western blotting using anti-FLAG or anti-ATG101. Western blot band quantification, ATG101 levels/HA-Ub levels from the immuno-complex subsequent to immunoprecipitation, is presented on the graph. Error bars indicate the mean ± SEM of three independent experiments. * *p* < 0.05.

**Figure 3 ijms-22-09182-f003:**
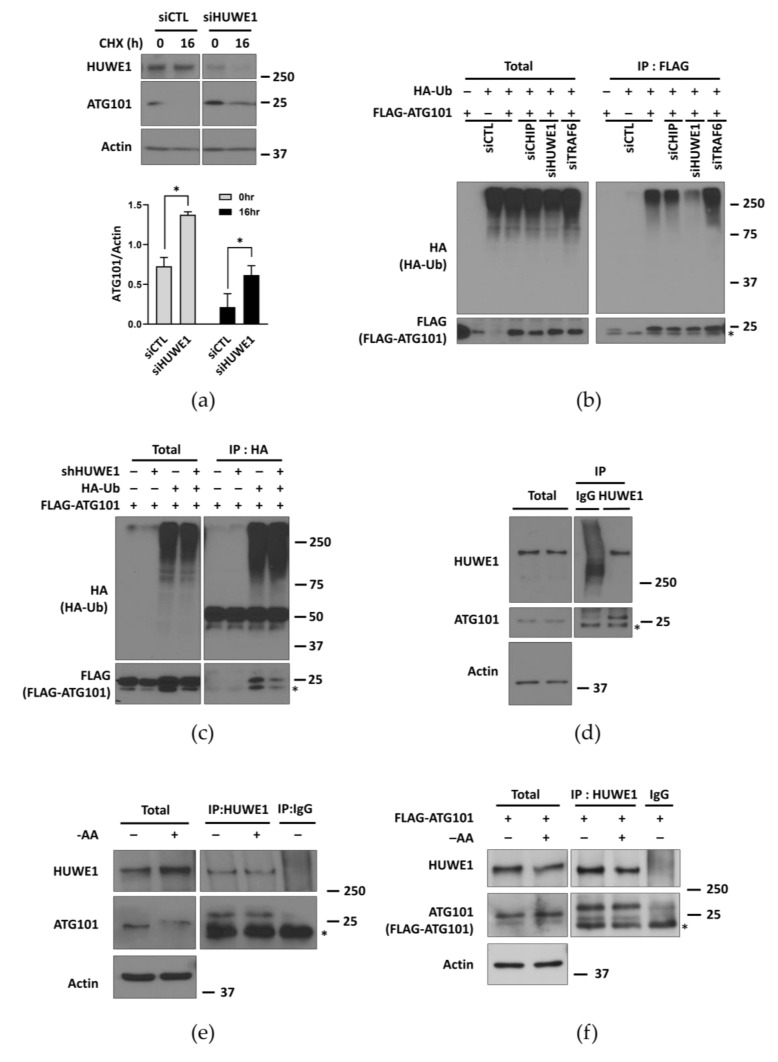
HUWE1 mediates ATG101 ubiquitination and degradation. (**a**) HUWE1 knockdown (KD) in MIA PaCa-2 cells increased ATG101 protein levels as measured by western blotting. MIA PaCa-2 cells were incubated with HUWE1 siRNA for 24 h and then incubated with CHX (10 μg/mL) at the time indicated. Cell lysates were analyzed by western blotting using the indicated antibodies. Quantification and statistical analysis were as in Figure 2a. (**b**) HUWE1 KD reduced ATG101 ubiquitination. HEK293T cells were transfected with siRNAs targeting ATG101 E3 ligase candidates (siCTL, siCHIP, siHUWE1, siTRAF6) for 24 h, then co-transfected with HA-Ub and FLAG-ATG101 for 24 h. FLAG-ATG101 was immunoprecipitated with anti-FLAG. Immunoprecipitates were analyzed by western blot using anti-HA for determining ATG101. * non-specific bands. (**c**) HUWE1 KD reduced ATG101 ubiquitination. shCTL or shHUWE1 HEK293T cells were co-transfected with HA-Ub and FLAG-ATG101. HA-ubiquitin was immunoprecipitated with anti-HA, then the immunoprecipitated complexes were analyzed for ATG101 ubiquitination by western blotting using anti-FLAG. * non-specific bands. (**d**,**e**) Co-immunoprecipitation demonstrates a possible physical interaction between HUWE1 and ATG101 in MIA PaCa-2 cells. Endogenous HUWE1 was immunoprecipitated using anti-HUWE1 and the immunoprecipitates were analyzed by western blot using anti-ATG101. Cells were cultured in (**d**) basal condition and (**e**) starvation condition for incubating cells with amino-acid medium –AA) for 2 h. * non-specific bands. (**f**) A physical interaction between HUWE1 and ectopic expressed ATG101 in both basal and nutrient deprivation. MIA PaCa-2 cells were transfected with FLAG-ATG101 for 48 h. Cellular starvation, immunoprecipitation, and subsequent western blot analysis were as described in Figure 3d,e. * non-specific bands.

**Figure 4 ijms-22-09182-f004:**
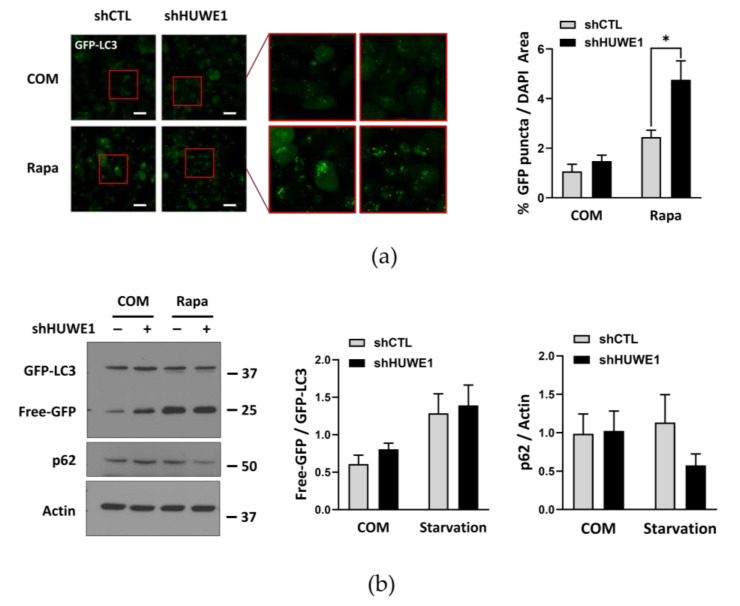
HUWE1 suppresses autophagy by regulating ATG101 and WIPI2 levels (**a**) HUWE1 KD enhanced GFP-LC3 puncta formation in MIA PaCa-2 cells. shCTL or shHUWE1 MIA PaCa-2 cells stably expressing GFP-LC3 were plated overnight, and subsequently incubated with rapamycin (0.5 μM) for 4 h prior to imaging. Cell imaging, quantification and statistical analysis were as described in Figure 1d. * *p* < 0.05. (**b**) HUWE1 KD showed enhancement of autophagy activity as indicated by western blotting for autophagy marker proteins. Cell lysates from (a) were analyzed by western blot using the indicated antibodies for autophagy marker proteins. Quantification and statistical analysis were as in Figure 1c. (**c**) ATG101 KD reversed the increase in GFP-LC3 puncta formation induced by shHUWE1 cells. shCTL or shHUWE1 MIA PaCa-2 cells stably expressing GFP-LC3 were transfected with ATG101 siRNA for 48 h then incubated with rapamycin (0.5 μM) for 4 h. Cell imaging, quantification and statistical analysis were as in Figure 1d. Scale bar: 20 μm. Error bars indicate the mean ± SEM of three independent experiments. ** *p* < 0.01; *** *p* < 0.001. (**d**) Both ATG101 and WIPI2 double KD reversed the increase in autophagy flux induced by shHUWE1. shHUWE1 HeLa cells stably expressing mCherry-GFP-LC3 were transfected WIPI2 and/or ATG101 siRNA for 48 h, followed by incubation in amino acid-deprived medium (–AA) for 4 h. Cells were imaged under starvation conditions as in (c). Scale bar: 20 μm. mCherry puncta area (%) was normalized to Hoechst-stained area and is presented as a percentage on the quantification graph. Error bars indicate the mean ± SEM of three independent experiments. ** *p* < 0.01 (**e**) A physical interaction between WIPI2 and ATG101. HEK293T cells were co-transfected with GFP-WIPI2 and FLAG-ATG101 and incubated for 24–48 h in the treatment with vehicle or rapamycin (0.5 μM) for 4 h prior to harvesting. GFP-WIPI2 was immunoprecipitated with anti-GFP, then the immunoprecipitated complexes were analyzed for interaction between ATG101 and WIPI2 by western blotting using anti-FLAG. (**f**) Removal of the ATG101 C-terminus domain (ΔC) reduced the interaction between ATG101 and WIPI2. HEK293T cells were co-transfected with GFP-WIPI2 and FLAG-ATG101 FL or ΔC for 24–48 h and incubated for 4 h in complete medium (COM) or amino-acid deprived medium (–AA) for starvation. FLAG-ATG101 was immunoprecipitated with anti-FLAG, then immunoprecipitated, analyzed by western blotting using anti-WIPI2.

**Figure 5 ijms-22-09182-f005:**
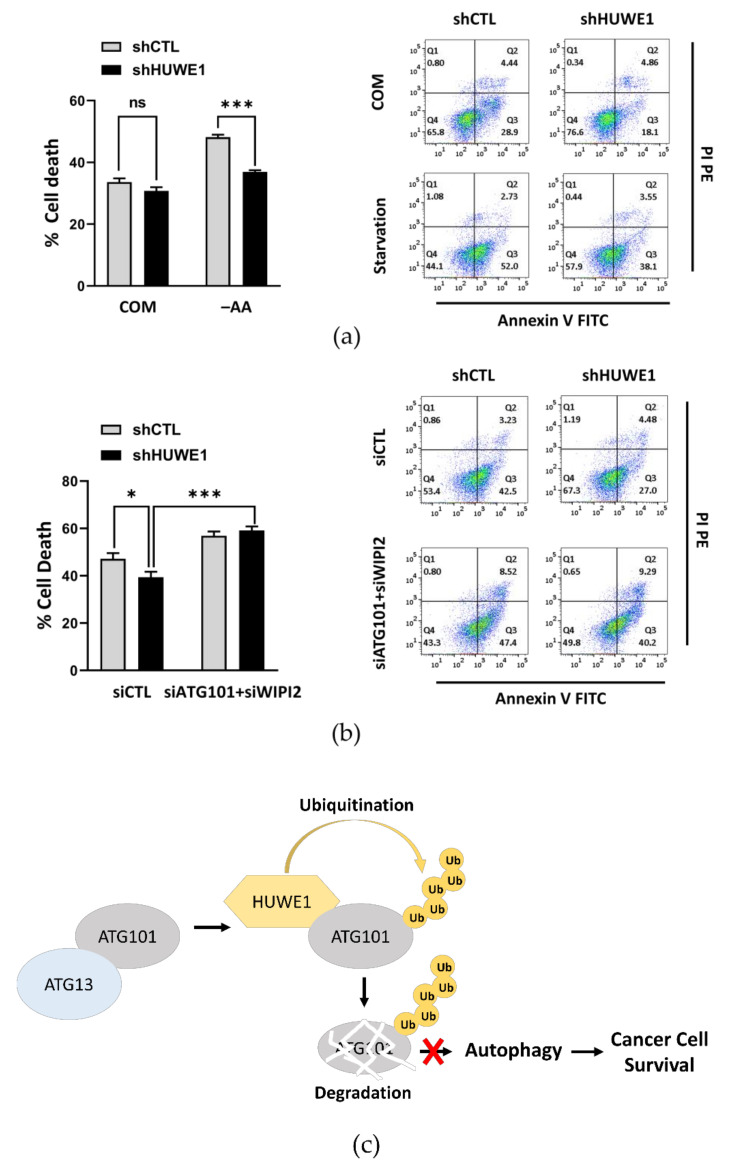
ATG101-mediated autophagy supported while HUWE1-catalyzed ubiquitination of ATG101 reduced cancer cell survival. (**a**) Knockdown of HUWE1 reduced the apoptotic death of cancer cells under nutrient starvation as assessed by Annexin V/propidium iodide (PI) staining and flow cytometry. shCTL or shHUWE1 MIA PaCa-2 cells were plated overnight and incubated in amino acid deprivation (–AA) medium for 48 h. Error bars indicate the mean ± SEM of three independent experiments. *** *p* < 0.001. (**b**) Additional knockdown of ATG101 and/or WIPI2 further enhanced cell death in HUWE1 knockdown cells. shCTL or shHUWE1 MIA PaCa-2 cells were transfected with WIPI2 and/or ATG101 siRNA for 24 h and subsequently incubated in HBSS for 24 h. Cell death was analyzed as in (**a**). Error bars indicate the mean ± SEM of three independent experiments. * *p* < 0.05; *** *p* < 0.001. (**c**) Schematic diagram of HUWE1 targeting ATG101 for ubiquitination and degradation, promoting cancer cell death via blocking autophagy activation under metabolic stress.

## Data Availability

Data is contained within the article and Appendix A.

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
