# Peer review of "ATG101 Degradation by HUWE1-Mediated Ubiquitination Impairs Autophagy and Reduces Survival in Cancer Cells"

_ijms, 2021, doi:10.3390/ijms22179182_

Round 1

Reviewer 1 Report

The paper presented by Lee and collaborators described the role of ATG101 ubiquitination, a partner in the ULK1 complex, by HUWE1 in autophagy and cancer cell survival. The team already described that ULK1 is ubiquitinated by NEDD4L that this inhibits autophagy pathway along with modulation of glutamine metabolism. The work is well presented, but even if the quality of the experiments performed is mostly good and especially the rescue experiment in figure 5b, however there are some experiments that need to be performed again or included. Moreover, defining 1/if the ubiquitination of ATG101 is different when autophagy is activated, 2/if this modification impact on the ULK1 complex and or/ interaction with downstream partners, may be addressed.

Other concerns are as follow:

  • The authors compared CRISPR KO cells to WT cells. The KO were obtained upon infection, and it’s admitted that infection can modify cell metabolism and other processes. Based on this, experiments conducted with KO cells may be performed again and compared with a CRISPR Ctl (unrelevant sgRNA) cell line.
  • ATG101 is very important for the initiation of the autophagy process as it is working with ULK1. It is not so evident to understand why the authors address the impact of ATG101 KO specifically on mitophagy, except to show that by modulating autophagy initiation you inhibit both the non-selective and selective autophagy, and that Huwe1 has already been shown to impact mitophagy process. If the authors want to keep this part, probably they should present it in supplementary figure and as a read-out of the global autophagy process inhibition. To continue on this aspect, the data presented in figure 1e are not very consistent, because it’s not possible to detect mitophagy in HeLa, when they are not overexpressing parkin, as soon as 4h after CCCP treatment. Finally, why is there no the staining of mitochondria with mitotracker red in the KO cells?
  • In figure 3, the authors analyzed the ATG101 ubiquitination by HUWE1 in basal and CCCP treated conditions. Starvation or HBSS treatment may be more appropriate, as the authors are interested in general autophagy regulation of ATG101, and not only the selective form of autophagy, mitophagy.
  • One of the important things to look at when studying autophagy is the flux (addressed in figure 4d). This has also to be done in the characterization of the KO or sh cells (figure 1c and figure 4b) by WB. The blot of LC3 is missing in figure 4b.
  • In figure 4c and 4d, the authors analyzed LC3 puncta, but the results regarding ATG101 KD in treated condition are not consistent between both experiments. This has to be performed again to show that the silencing of ATG101 is inhibiting autophagy, as expected.
  • The authors were also interested in the role of ATG101 in cancer cell death and proliferation. Clonogenic assay, which combined both cell death and proliferation analysis, would help to go deeper in the role of ATG101 modulation in those processes.
  • Globally, the quantification of the WB is not fitting with representative images (as an example in figure 2e,f).
  • The quality of the pictures as to be improved, in all the figure concerned.

Author Response

We provide a point-by-point response to your valuable comments in the attached file.

Reviewer 2 Report

The submitted manuscript is an original article entitled "ATG101 degradation by HUWE1-mediated ubiquitination impairs autophagy and reduces survival in cancer cells." is a manuscript by Lee et al. 

Minor Point:

Please add a molecular weight marker to all Western Blot data figures.

Major Point:

Because HUWE1 has a central role in several pathways, it would help to support the idea of the authors if they could present some data based on conditional or point-mutated versions of HUWE1.

Author Response

(The authors gave the same response as above.)

Reviewer 3 Report

The authors should improve the manuscript to describe each experiment and result more in detail and appropriately.

There are many experients and each result that have not been cited or mentioned in the manuscript.

Author Response

(The authors gave the same response as above.)

Round 2

Reviewer 2 Report

The authors have provided a sufficiently revised manuscript. The manuscript is acceptable for publication in its current form.